# Production and Characterization of Snacks Utilizing Composite Flour from Unripe Plantain (*Musa paradisiaca*), Breadfruit (*Artocarpus altilis*), and Cinnamon (*Cinnamomum venum*)

**DOI:** 10.3390/foods13060852

**Published:** 2024-03-11

**Authors:** Mojisola Olanike Adegunwa, Busayo Olabisi Ogungbesan, Olasunkanmi Abdulganiu Adekoya, Ebunoluwa Eniola Akinloye, Oluwadolapo Daniel Idowu, Oladeji Emmanuel Alamu

**Affiliations:** 1Department of Hospitality and Tourism, Federal University of Agriculture, Abeokuta 111101, Ogun State, Nigeria; adegunwamo@funaab.edu.ng (M.O.A.); ogungbesanbo@funaab.edu.ng (B.O.O.); adekoyaganiu1@gmail.com (O.A.A.); akinloyeee.17@student.edu.ng (E.E.A.); ayomikunenihorlar@gmail.com (O.D.I.); 2Food and Nutrition Sciences Laboratory, International Institute of Tropical Agriculture, Southern Africa Research and Administration Hub Campus, Lusaka 10101, Zambia; 3Food and Nutrition Sciences Laboratory, International Institute of Tropical Agriculture, PMB 5320, Oyo Road, Ibadan 200001, Oyo State, Nigeria

**Keywords:** composite flour, cookies, quality evaluation, anti-nutritional properties, sensory properties

## Abstract

This study aimed to assess the characteristics of flour and cookies produced from composite plantain, breadfruit, and cinnamon blends. Six samples were created by combining the flours (Plantain, Breadfruit, and Cinnamon Composite (PBC)) in the following proportions: 100:0:0, 0:100:0, 65:30:5, 70:25:5, 75:20:5, and 80:15:5. The flours were subjected to proximate, functional, and color analyses using standard methods. The results obtained included moisture content (7.13–9.56%), ash (6.37–7.14%), protein (8.38–12.62%), fat (8.36–12.92%), crude fiber (6.54–7.52%), and carbohydrate (51.84–60.83%). The functional properties of the flour included bulk density (0.66–0.80 g/mol), water absorption capacity (1.91–2.87%), oil absorption capacity (0.88–1.91%), swelling power (3.89–5.30), solubility index (0.01–0.04%), dispersibility (48.55–66.05%), tannin content (1.71–3.98 mg/g), and phytate content (1.57–3.35 mg/g). The analyses revealed significant differences (*p* < 0.05) in the proximate composition of the cookie samples. The values ranged from 6.19 to 7.99% for moisture content, 16.06 to 24.75% for crude fat, 3.10 to 8.13% for crude ash, 3.62 to 7.15% for crude fiber, 8.18 to 10.99% for crude protein, and 43.82 to 59.03% for carbohydrates. Substantial changes (*p* < 0.05) in color analysis and sensory qualities were observed in the cookie samples, and sample B (100% breadfruit flour) had the most appreciable color attributes. However, the cookies produced from 75:20:5 plantain–breadfruit–cinnamon flour had the highest overall acceptability along with improved nutritional properties. These findings suggest that adding breadfruit and cinnamon flour to cookies could improve their nutritional content, particularly in ash, fiber, fat, and protein.

## 1. Introduction

As consumer demand for nutritious foods and health benefits increases, new product development is critical in the food industry [1]. Food fortification has a long history of use in some countries and has effectively controlled micronutrient deficiencies. Fortified flours can be an essential source of bioactive substances that reduce the risk of various deficiencies and provide health benefits [2].

Plantain plants are monocotyledonous perennial crops that provide significant food worldwide in tropical and subtropical regions [3,4]. It is an important food crop in Sub-Saharan Africa, providing many people with nutrition and household income worldwide. Plantain flour is a product of processed, unripe plantains. They are widely consumed since they are economical and highly nutritional. Most raw materials used in home and flour manufacturing come from large amounts of under-ripe bananas or postharvest loss [5,6]. Due to their limited shelf life, plantains are fruits that spoil quickly. Because of this, they are frequently processed into long-lasting goods, including chips, noodles, bread, cookies, and flour [7,8,9].

Breadfruit, *Artocarpus altilis*, emerges as an exceptionally versatile and nutrient-rich fruit, serving as a staple food that can significantly enhance human health and environmental well-being [10,11]. Breadfruit is a versatile potato substitute in dishes, accommodating diverse cooking methods and even technological applications in producing cookies, breads, and flour [12,13]. Breadfruit’s glycemic index is notably lower than many everyday staples, including wheat, cassava, yam, and potatoes [14]. Beyond its culinary appeal, breadfruit is low in fat and cholesterol, high in nutrients and complex carbohydrates, and boasts an extensive array of amino acids and vitamins [15,16].

Cinnamon is an evergreen aromatic tree from the *Lauraceae* family, which is one of the most widely studied flowering groups, with over 250 different species [17]. It is an important spice used worldwide, as researchers from the Malaysian and United States (US) departments of agriculture have reported. These studies have identified several therapeutic effects of cinnamon, such as its antioxidant properties and its positive impact on conditions such as diabetes and neurological, microbial, and cardiovascular diseases, attributed to its bioactive components [18]. Cinnamon contains several essential nutrients, including 38% of the recommended daily manganese intake and 10% of the daily intake for iron, dietary fiber, and calcium [19]. Cinnamon has also been traditionally used for its antibacterial, antifungal, antimicrobial, astringent, and anticlotting properties and is often included in tea as a remedy for nausea during pregnancy or to decrease post-delivery hemorrhage [20].

## 2. Materials and Methods

### 2.1. Source of Materials

The plantains, breadfruit, and cinnamon were procured from the Kuto market in Abeokuta, Ogun State, Nigeria. Also, all other ingredients, including margarine, granulated sugar, baking powder, salt, vanilla extract, powdered milk, and fresh eggs, were shopped from a local supermarket.

### 2.2. Plantain Flour Processing

The process of making flour from mature, unripe plantains involved sorting, weighing, and cleaning them thoroughly. The plantains were then peeled with a knife to extract the pulps and sliced into pieces around 5 mm in diameter. The sliced plantains were then oven-dried at 60 °C for 24 h, following the method described by Adegunwa et al. [21]. After the drying process, the plantain slices were hammer-milled into flour and sieved using a 0.25 mm sieve. Finally, the finished flour was packaged and sealed in Ziploc bags.

### 2.3. Cinnamon Powder Processing

The cinnamon was grated and blended into cinnamon powder, and the resultant flour was sieved [21].

### 2.4. Breadfruit Flour Processing

The breadfruit flour was prepared from fresh fruits of breadfruit by washing the fruits under running water first to remove dirt. Each of the fresh fruits was peeled and sliced into chips. The chips were then oven-dried in a cabinet oven dryer at 60 °C for 24 h. The dried chips were then milled and sieved through a 0.25 mm sieve and packed in Ziplock bags for further use [10].

#### Production of Cookies 

Adegunwa et al. [22] described the recipe and method that were used. The ingredients are unripe plantain flour, breadfruit, and cinnamon (in varying proportions), margarine (40 g), sugar (30 g), baking powder (2 g), salt (0.5 g), vanilla flavor (0.5 g), milk (10 g), and an egg (1 piece). Combine the margarine and sugar in a mixing bowl, then beat at medium speed until fluffy. Add one whole egg and milk powder to the cream, mix for about 30 min, and slowly introduce the composite flour, baking powder, salt, and vanilla flavor. The dough should achieve a thick consistency on a floured rolling board. Cut the dough into appealing shapes with cookie cutters and place on well-greased baking trays. Bake in an oven at 180 °C for about 20 min until pale brown, then cool to room temperature and pack in polyethylene bags for analysis. Prepare the cookies in duplicate.

### 2.5. Formulation of Composite Flour from Plantain, Breadfruit, and Cinnamon

About 100 g of flour from plantain (P), breadfruit (B), and cinnamon (C) in various proportions (A—100:0:0; B—0:100:0; C—65:30:5; D—70:25:5; E—75:20:5; F—80:15:5) were thoroughly mixed, packed, and analyzed using standard laboratory procedures (Table 1).

### 2.6. Proximate Composition Determination

The Association of Official and Analytical Chemists [23] provided an official analytical method for determining the approximate composition of protein, ash, moisture content, crude fat, crude fiber, and carbohydrates.

### 2.7. Functional Property Determination 

#### 2.7.1. Bulk Density Measurements

The bulk density was determined using method of Akpapunam and Markakis [24], as described by Adegunwa et al. [25]. A 10 g sample was measured into a graduated measuring cylinder with a capacity of 50 mL. The sample was carefully packed by repeatedly pounding the cylinder on a bench top until there was no more volume reduction. The bulk density was calculated using the recorded volume of the compacted sample and the following formula:$$Bulk\;density\left(\frac{g}{mg}or\frac{g}{{cm}^{3}}\right)=\frac{Weight\;of\;sample}{Volume\;of\;the\;sample\;after\;tapping}$$

#### 2.7.2. Water Absorption Capacity Measurements

This was determined using Sosulski’s [26] methodology. We added 15 mL of distilled water to 1 g of sample in a previously weighed centrifuge tube. The tube contents were violently agitated for 2 min on a flask Gallenkamp shaker before being centrifuged at 4000 rpm for 20 min with a SORVALL GLC-1 centrifuge (Model 06470, ALT, San Diego, CA 92130, USA). The transparent liquid portion was discarded, and the centrifuge tube was weighed alongside the solid residue. The sample’s water content was estimated using the difference method and reported as the weight of water absorbed by 100 g of dehydrated flour.

#### 2.7.3. Oil Absorption Capacity Determination

Sosulski’s method [26] was used to determine the oil absorption capacity. A one-gram flour sample was mixed with 10 mL of oil and left to sit at room temperature for 30 min before being centrifuged for 30 min at 3000 rpm. The clear supernatant was decanted. The oil absorption was calculated using the percentage of oil bound per gram of flour.

#### 2.7.4. Swelling Power and Solubility Index Measurements 

The method given by [27] was employed to determine the swelling power and solubility index. The procedure entailed measuring 1 g of the specimen and placing it into a 50 mL centrifuge tube. Approximately 50 mL of distilled water was added, and the mix was stirred delicately. The slurry underwent thermal treatment in a water bath at 80 °C for 15 min. While being heated, the slurry was agitated delicately to avoid the formation of flour clumps. The paste tube was centrifuged at 3000 rpm for 10 min after a time interval of 15 min. The liquid portion was poured off promptly following the process of centrifugation. The sediment’s weight was measured and documented. The sediment gel’s moisture was measured to obtain the gel’s dry matter content.

### 2.8. Pasting Property Determination

The samples’ pasting properties were assessed using the methodology described by Alamu et al. [28]. A Rapid Visco Analyzer (RVA, model 3C; Newport Scientific Pty Ltd., Sydney, NSW, Australia) was used. Peak viscosity, holding strength, breakdown, final viscosity, setback, peak time, and pasting temperature were measured with a computer (Thermocline for Windows Software 11.2; Newport Scientific).

The analysis used analytical-grade chemicals, and each analysis was carried out twice.

### 2.9. Phytate Content Determination

The amount of phytate in the given samples was determined using the AOAC [23] method. Firstly, the phytate was extracted using a 0.5 mol/L (HNO_3_) solution and then digested with 0.5 mL of perchloric acid (HClO_4_). The digested sample was further mixed with 25 mL of distilled water. Then, 2.5 mL of this mixed extract was taken and blended with 2.5 mL of nitric acid. After that, the phytic phosphorus in the mixture reacted with 2.5 mL of vanadium molybdate reagent to form a yellow-orange complex. The absorbance (OD) was measured at 460 nm with a spectrophotometer (P7 UV/Vis spectrophotometer, Shanghai Mapada Instrument Coy. Ltd, Shangai City, China). Finally, the phytate content was calculated using a standard concentration of 2 mg phytic acid and a reagent blank treated identically to the sample above.

### 2.10. Tannin Content Determination

The AOAC [23] technique was used to determine the tannin concentration. The sample (5 g) was placed into 50 cc of distilled water and agitated. The mixture was left undisturbed for 30 min at a temperature of 28 °C before being passed through a Whatman grade 4 filter paper. A total of 2 mL of the extract was transferred into a 50 mL volumetric flask. Similarly, a separate volumetric flask was used to prepare a standard solution consisting of 2 mL of a tannic acid solution with a concentration of 0.1 mg/mL and 2 mL of distilled water. Next, 2.5 mL of a solution containing the maximum amount of sodium carbonate (Na_2_CO_3_) it could dissolve and 1 mL of Folin-C reagent were added to each flask. The capacity of each flask was then increased to 50 mL and thoroughly mixed. Following 1½ hours of standing, the sample underwent filtration using a Whatman grade 4 filter paper. Subsequently, the absorbance was quantified at 760 nm relative to a reagent blank.

### 2.11. Color Property Measurements

The composite flour’s color properties (L*, a*, b* values) were measured using the Minolta CR-400 Chroma Meter (Minolta Corp., Osaka, Japan) following the CIE method [23]. The colorimeter uses the CIE (Commission Internationale de l’Eclairage) L*, a*, b* color scheme to measure color. To start, the instrument was standardized with a piece of Business Xerox 80 g/m^2^ white paper with whiteness D65 and 136 CIE. Then, multiple measurements were taken at several points on the sample to obtain accurate readings.

### 2.12. Sensory Evaluation

The snacks’ sensory attributes were obtained using simple hedonic tests described by Adegunwa et al. [21]. This was carried out using a 20-member panel consisting of students from the department who were familiar with the sensory attributes of the snacks. Each panelist was asked to score each attribute on a 9-point hedonic scale, where 1 and 9 represented dislike extremely and like extremely, respectively. The evaluated attributes included color, taste, flavor, texture, appearance, crispness, and overall acceptability.

### 2.13. Statistical Analysis

The data from the chemical analysis were statistically analyzed using ANOVA, and the mean was separated using Duncan. The Statistical Package for Social Scientists (SPSS) software (version 13) was used to compute the Multiple Range Test at a *p*-value of <0.05. All samples were analyzed in duplicate. 

## 3. Results and Discussion

### 3.1. Proximate Composition of Composite Flour from Plantain, Breadfruit, and Cinnamon

Table 2 shows the composition of the mixed flour. The moisture content of the samples ranged from 7.13 to 9.56%. The highest moisture content was found in the 100% plantain flour and the lowest in the PBC (80:15:5) flour. The moisture content decreased with the addition of breadfruit and cinnamon flour. This study’s moisture content is lower than that reported by Adegunwa et al. [25], which was 10.75%. The moisture content of flour affects its shelf life. The low moisture content recorded for all sample blends suggests that the samples are of good quality for storage [29]. The composite flour’s moisture content was less than the SON [30] (Standard Organization of Nigeria) recommendation of 10% for flour sample storage. The protein, ash, and crude fiber contents of the PBC (80:15:5) flour increased as the amount of breadfruit and cinnamon flour inclusion increased. This is because breadfruit and cinnamon flour have higher levels of these nutrients than 100% plantain flour. Eleazu et al. [31] also reported that unripe plantain (UP) flour has low crude protein content, and a 100% UP flour diet cannot meet the protein requirement in adults.

All the samples had an ash content ranging from 6.37% to 7.94%, with the highest value observed in PBC (80:15:5). The high ash content of the inclusion flour indicates that it could be a good source of minerals, thereby improving the mineral content of the composite flour. The ash concentrations of the flour blends exhibit a statistically significant difference (*p* < 0.05). Ash is the inorganic residue left behind when water and organic matter are removed by heating in the presence of an oxidizing agent. The low quantity of ash obtained in 100% plantain flour indicates its low mineral content. The increase in ash and mineral content with increased PBC (80:15:5) inclusion indicates nutrient enhancement caused by the addition of breadfruit and cinnamon flour. This agrees with past reports by Eleazu et al. [31] and Adegunwa et al. [25].

The fiber content of the plantain component of the flour blend was lower than that of the composite flour because of the inclusion of breadfruit and cinnamon flour. PBC (80:15:5) had the highest value (7.52), while PBC (65:30:5) had the lowest fiber content (6.54%). The protein content varied significantly from PBC (65:30:5) (8.38%) to PBC (80:15:5) (12.62%), and the fat content of the flours ranged from PBC (65:30:5) (8.36%) to PBC (80:15:5) (12.92%). The positive impact breadfruit had on the protein and fat contents of the composite flour suggests nutrient enrichment in the flour blends because of breadfruit’s high protein and fat content combined with the addition of cinnamon flour [32,33].

Crude fat indicates the amount of unrefined fatty substances present in a product. A higher crude fat content suggests that the composite flour may be a good source of fat-soluble vitamins. The carbohydrate content ranged between 51.84% and 60.83%, with the highest value recorded for the PBC (65:30:5) flour. The addition of breadfruit to the plantain flour caused an increase in the energy level of the flour blends. 

The chemical compositions of the flour obtained in this study reflect lower moisture, fats, and crude fiber contents but higher protein, ash, carbohydrate, and energy values when compared to those reported by Adegunwa et al. [25,34]. The higher levels of protein, fat, crude fiber, and ash, along with the lower carbohydrate content, in the composite flour confer a higher nutritional advantage over both 100% plantain flour and 100% breadfruit flour. This is especially good for individuals with health conditions requiring foods rich in protein and fiber and low in carbohydrates [35]. The PBC (80:15:5) flour blend could be a useful source of functional dietary fiber and resistant starch. It exhibits elevated levels of total and soluble fiber, improved abilities to retain water and oil, and increased capacity to expand; it also promotes the proliferation of probiotic organisms [21,36]. Therefore, its inclusion in composite flour for cookies can potentially deliver safe foods for patients with hypertension and diabetes.

The results of the anti-nutritional factors of tannin and phytate showed that PBC (80:15:5) had the highest tannin content (3.98 ± 0.92 mg/g), while 100% breadfruit had the lowest tannin content (1.71 ± 0.00 mg/g). Tannins have both beneficial and detrimental effects on health. They possess antioxidant qualities, which can help prevent specific diseases. However, high levels of dietary tannins can interfere with nutrient absorption and negatively affect protein digestion. The phytate content was highest in PBC (80:15:5) (3.35 ± 0.46 mg/g) and lowest in PBC (65:30:5) (1.57 ± 0.99 mg/g). The phytate content significantly affected the nutritional quality of these samples. The highest phytic acid value, recorded for 80:15:5, was expected due to the higher percentage of plantain flour in the blend. Thus, it could be inferred that the lower the plantain flour in the blend, the lower the phytate level. Phytate is an inhibitor of the absorption of minerals in the body [37,38].

### 3.2. Functional Properties of Composite Flour from Plantain, Breadfruit, and Cinnamon

The functional properties of the plantain–breadfruit–cinnamon flour blends were analyzed, and the results are shown in Table 3. These properties determine the usage of food materials for various products. Bulk density, which measures the amount of space occupied by the flour, ranged from 0.66 ± 0.00 to 0.80 ± 0.00 g/mL. The flour with the highest bulk density was PBC (80:15:5), while 100% breadfruit flour had the lowest value. A significant difference (*p* < 0.05) in bulk density was observed. A high bulk density makes the flour an appropriate thickener in food products and for food preparation since it helps reduce paste thickness, which is an essential factor in convalescent and child feeding [39,40].

Water absorption capacity indicates the amount of space the flour takes up when it expands in excessive water. This measurement indicates gelatinization. There was a significant difference in the water absorption index of the composite flour blends (*p* < 0.05). PBC (65:30:5) had the highest mean value of 2.87 ± 0.01, while PBC (70:25:5) had the lowest mean value of 1.91 ± 0.02. The water-binding capacity values ranged from 19.00 ± 6.20 to 66.80 ± 20.00. The water-binding capacity of plantain–breadfruit flour increased with breadfruit substitution due to increased protein and fat content. This improves the reconstitution and textural ability obtained from the plantain flour [35,41].

The oil absorption capacities of the flour blends were generally high, ranging from 0.88 to 1.91 g/mL. Flour’s ability to absorb oil is vital, as oil acts as a flavor retainer and improves mouthfeel. PBC (80:15:5) had the highest oil absorption capacity value of 1.91 g/mL, while 100% plantain flour had the lowest value of 0.88 g/mL. The rise in oil absorption capacity with the addition of breadfruit flour and cinnamon flour could be due to the differences in the presence of a non-polar side chain, which might potentially bind with the hydrocarbon side chain of the oil present in the flours [42,43].

The samples did not exhibit statistically significant differences (*p* > 0.05) in their swelling power at a temperature of 100 °C. The solubility index of the samples at 100 °C did not show any statistically significant change (*p* > 0.05). The dispersibility results show that 100% plantain flour had the highest value (66.05) of dispersibility, while 100% breadfruit flour had the lowest value of 48.55.

#### 3.2.1. Color and Anti-Nutrient Properties of Flour Produced from Plantain, Breadfruit, and Cinnamon

Table 4 displays the analysis of the color and anti-nutritional properties of composite flour made from plantain, breadfruit, and cinnamon. We found significant differences (*p* < 0.05) in the color properties (L*, a*, b*, and ΔE*) of the composite flour. The highest lightness (93.36 ± 0.49 RVU) was recorded for 100% breadfruit flour, whereas PBC (75:20:5) had the least lightness (84.63 ± 0.00 RVU). However, PBC (75:20:5) recorded the highest redness value (2.42 ± 0.00 RVU), and 100% breadfruit flour had the lowest (0.70 ± 0.03 RVU). Moreover, 100% breadfruit flour had the highest yellowish value (19.47 ± 0.21 RVU), and 100% plantain flour had the lowest (15.88 ± 0.00 RVU). These color attributes can affect consumer preferences, as color often influences taste and quality. Thus, controlling and understanding these attributes is crucial in food production and marketing. These color measurements can also help maintain consistency and quality control in food manufacturing [44,45]. 

The drying process significantly impacted increased lightness due to the loss of moisture and non-enzymatic Maillard browning, which occurred under the conditions prevailing during the drying process, favoring color change. Fadimu et al. [46] reported a similar result of L > 50 for plantain flour.

#### 3.2.2. Pasting Properties of Flours Produced from Plantain, Breadfruit, and Cinnamon

Table 5 displays the pasting properties of the flours. Pasting properties refer to the changes in food due to heat application in the presence of water. These changes affect the food product’s texture, digestibility, and end-use. Peak viscosity is the stage at which gelatinized starch attains its highest viscosity when heated in water. This indicates starch’s water-binding ability. A decline usually follows the phenomenon of peak viscosity, reaching a minimum level known as trough viscosity. This decline occurs due to the rupture and release of starch granules when exposed to high temperatures and shear forces. Of all the samples, PBC (80:15:5) had the highest peak viscosity value of 500.98 ± 9.62 RVU, while 100% breadfruit flour had the lowest with 331.12 ± 0.85 RVU. Starches that exhibit high peak viscosity are likely to have high breakdown values, leading to weak gels. Such gels are most likely to disintegrate under shear and heat. The high peak viscosity value of 100% plantain flour has a processing advantage, making it a good material for producing stiff dough products with better palatability and water-binding capacity [38,47].

The disruption of granules results in a decrease in paste viscosity, termed trough viscosity. The trough viscosity, or holding strength, is the minimum viscosity reached at the end of a holding duration at the heating temperature. PBC (80:15:5) had the highest trough viscosity value of 367.87 ± 0.74, while 100% plantain flour had the lowest value of 27.76 ± 1.27 RVU, indicating its susceptibility to degradation throughout the cooking process. 

Breakdown viscosity is a measure of the resistance to heat and shear. The results show that PBC (65:30:5) had the highest value (724.80 ± 64.35 RVU), while 100% breadfruit flour had the lowest value (49.20 ± 1.44 RVU). A more excellent breakdown is considered a sign of enhanced palatability. 

Final viscosity is the predominant parameter used to assess the quality of a sample based on its starch content. Of all the samples examined, PBC (65:30:5) had the highest value of 570.81 ± 5.23 RVU, and 100% plantain flour had the lowest value of 472.25 ± 0.57 RVU. The rise in viscosity from the lowest to the ultimate value is known as the setback and has been linked to the texture of different final goods. PBC (75:20:5) had the highest setback viscosity value of 248.56 ± 3.52 RVU, while 100% plantain flour had the lowest value of 210.99 ± 1.13 RVU. Since PBC (75:20:5) recorded the highest setback viscosity value, it can remain undisrupted when subjected to a long period of constant high temperature. 

The time required to reach peak viscosity may be used to predict the stability ratio and, consequently, the resistance of a starch paste to break down. Our results show that 100% breadfruit flour had the highest value of 5.40 ± 0.99 min, while 100% plantain flour had the lowest value of 5.16 ± 0.57 min. Finally, the pasting temperature results show that PBC (80:15:5) had the highest value of 88.24 ± 0.28 °C, while 100% breadfruit flour had the lowest value of 84.54 ± 0.18 °C [48,49].

#### 3.2.3. Proximate Composition of Cookies Produced from Plantain, Breadfruit, and Cinnamon Flour

Table 6 presents the results of the proximate composition analysis of cookies made from composite flour of plantain, breadfruit, and cinnamon. The flour samples’ moisture levels ranged from 6.19% to 7.99%. PBC (65:30:5) had the highest moisture content (7.99%), while 100% breadfruit had the lowest (6.19%). Low moisture content contributes to the longer shelf life of food products. Moisture levels above 14% facilitate microorganisms’ growth, causing spoilage and decreased nutritional value. Therefore, food products with reduced moisture content have a longer shelf life. The samples had a notably low moisture level, which could enhance their capacity for long-term storage. High moisture content can have an adverse effect on storage stability, so the cookie sample with the highest moisture content may have a reduced shelf life compared to other samples with lower moisture content. The samples’ crude fat content analysis revealed a significant increase in fortified cookies at each fortification level, ranging from 16.06% to 24.75%. The highest value was in the PBC (75:20:5) flour. Fat plays a vital role in the shelf life of food products, but high-fat-containing food products deteriorate quickly. Also, too much fat is very undesirable in food items [50]. Although no shelf-life studies have been conducted, the cookie samples obtained during this research may have a good and reasonable shelf life, with the cookie containing the highest breadfruit flour having the highest possibility. 

The ash content of the cookies ranged from 3.10% to 8.13% for all samples, with the highest value in PBC (75:20:5) and the lowest (3.10%) in 100% breadfruit flour. Ash content indicates mineral richness, so samples with a higher ash content are expected to possess a comparatively higher mineral content. The increase in ash content with a corresponding increase in breadfruit flour and cinnamon flour substitution in the composite flour was not surprising. Breadfruit has been reported to contain a higher ash content than wheat. This could mean that cookies with breadfruit flour supplementation could help enhance many people’s mineral intake. The ash content of a food sample indicates its total inorganic mineral content, including macro- and microelements. This could benefit numerous individuals by enhancing their nutritional intake, particularly in fulfilling their daily mineral needs [47]. 

The fortified cookies exhibited a range of crude fiber content, ranging from 3.62% to 7.15% across all samples. The highest value was found in PBC (75:20:5) (7.15%), while the lowest value of ash content was found in 100% breadfruit flour (3.62%). The cookies’ crude fiber content consistently rose as the fortification level rose. Breadfruit has a higher concentration of crude fiber than wheat flour, and these findings could provide a rationale for the results obtained from the various cookie samples. This assertion can be substantiated by the discoveries made by Mepba et al. [51]. Consequently, breadfruit-infused cookies have the potential to be well-embraced by a wide range of individuals and health institutions.

The addition of fortification to cookies led to a noteworthy rise in the amount of crude protein, with values ranging from 8.18% (PBC (70:25:5)) to 10.99% (100% breadfruit flour). The higher protein content in breadfruit flour than in the composite flour could have been responsible for the higher protein content in the fortified cookies. These results are similar to those obtained using wheat and African breadfruit composite flour in cookie production [52], as well as the results of Ukeyima et al. [53] for wheat, soy, and carrot flour blends and the results obtained by Adegunwa et al. [54] for plantain–groundnut–cinnamon composite flour. The results showed that the cookies could address the prevalence of protein–energy malnutrition (PEM) in most of our communities in developing countries with high PEM prevalence. 

The carbohydrate contents of the samples showed a significant reduction in carbohydrate content from 100% breadfruit flour to PBC (75:20:5). The 100% breadfruit flour exhibited the highest value of carbohydrates (59.03%), while PBC (75:20:5) exhibited the lowest value of carbohydrates (43.82%). This is similar to a report where high carbohydrate results were reported by Yusuf et al. [55] and Giwa and Abiodun [56]. Almasodi [57] reported a similar trend for ginger powder bakery products.

#### 3.2.4. Color Attributes of Cookies Produced from Plantain–Breadfruit–Cinnamon Flour

The color parameters showed a significant difference in the flour blends’ lightness, redness, yellowness, and color intensity, as shown in Table 7. All the flour samples showed moderate degrees of lightness (24.61–33.28), low degrees of redness (4.82–6.27), moderate–high degrees of yellowness (13.94–22.75), and moderately high color differences (15.69–24.87). The difference in lightness (L*) could be attributed to the effect of enzymatic browning, which occurred under the conditions prevailing during the drying process, favoring color change [46]. The drying process significantly impacted increased lightness, which could be attributed to the loss of moisture and non-enzymatic Maillard browning, which occurred under the conditions prevailing during the drying process, favoring color change [58,59].

### 3.3. Sensory Properties of the Cookies

The findings of the sensory evaluation for the cookies are presented in Table 8. The results indicate that PBC (65:30:5) had the highest score in terms of color at 6.85%, followed by PBC (75:20:5) at 6.71%, while the lowest score was seen for PBC (70:25:5) at 5.90%. Our study revealed that color scores increased with the inclusion of cinnamon flour. The taste evaluation showed that fresh cookies made from 100% plantain flour achieved the highest score of 7.33, followed by PBC (80:15:5) with 6.76, while 100% breadfruit flour had the lowest score (5.42). The taste rating decreased randomly with the increasing inclusion of cinnamon flour and breadfruit flour in the cookie formulation. The highest aroma score was given to the cookies made from 100% plantain flour (6.85), followed by PBC (80:15:5) (6.71), while the lowest score was for 100% breadfruit flour (5.66). The sensory analysis for crunchiness shows that PBC (65:30:5) had the highest score (7.28), followed by 100% plantain flour. The lowest score for crunchiness was obtained by PBC (70:25:5). The sensation of crispness is detected when food is chewed using the molars and is typically characterized by its hardness and ease of being broken down [60,61]. The sensory score for texture was highest for PBC (80:15:5), followed by 100% plantain flour, with 7.09 and 7.04, respectively. The lowest score for texture was for PBC (70:25:5), with 6.38. This study clearly shows that the texture score of the cookies increased with the inclusion of breadfruit flour and cinnamon flour. The cookies made from 100% plantain flour had the best taste and aroma. PBC (65:30:5) had the best crunchiness, while PBC (80:15:5) had the best texture. Surprisingly, the overall acceptability revealed that the cookies including PBC (75:20:5) were the most acceptable (7.52) (which could be due to their color and taste), followed by the cookies made from 100% plantain flour (7.33). PBC (70:25:5) had the lowest overall acceptability (6.25).

## 4. Conclusions

This study’s findings provide valuable information for developing food products and quality control. The results confirm that it is possible to produce flour with the right moisture content levels for a stable shelf life. These flours are rich in dietary fiber and essential nutrients and exhibit different functional behaviors that make them suitable for food production as ingredients. The inclusion of breadfruit and cinnamon substantially increased the protein, lipid, crude fiber, and carbohydrate levels of the composite flour and the resulting cookies. Adding breadfruit and cinnamon increased bulk density, water absorption capacity, and solubility. Producing cookies from composite plantain, breadfruit, and cinnamon flour blends for patients with hypertension and diabetes is feasible. The PBC (75:20:5) formulation produced the most acceptable cookies. The differences in pasting properties, color attributes, and anti-nutritional factors among the samples highlight the importance of selecting and processing these ingredients for various food applications. The use of composite flour would be economically significant in many developing countries, such as Nigeria and other African countries, promoting the use, utilization, and processing of local crops.

## Figures and Tables

**Table 1 foods-13-00852-t001:** Formulation of the composite flour.

Samples	Plantain Flour (P) (%)	Breadfruit Flour (B) (%)	Cinnamon Flour (C) (%)
A	100	0	0
B	0	100	0
C	65	30	5
D	70	25	5
E	75	20	5
F	80	15	5

**Table 2 foods-13-00852-t002:** Proximate composition and anti-nutritional properties of the composite flour (*n* = 12).

P:B:C Sample	Moisture Content (%)	Crude Fat (%)	Crude Ash (%)	Crude Fiber (%)	Crude Protein (%)	Carbohydrates (%)	Tannin mg/g	Phytate mg/g
100:0:0	9.56 ± 0.52 ^c^	10.98 ± 0.38 ^b^	6.37 ± 0.21 ^a^	6.62 ± 0.22 ^ab^	10.02 ± 0.54 ^b^	56.44 ± 0.04 ^c^	2.58 ± 0.00 ^b^	1.81 ± 0.14 ^a^
0:100:0	9.16 ± 0.49 ^c^	10.61 ± 0.36 ^b^	7.12 ± 0.24 ^b^	6.77 ± 0.23 ^abc^	9.70 ± 0.53 ^ab^	56.62 ± 0.07 ^s^	1.71 ± 0.00 ^a^	2.44 ± 0.09 ^b^
65:30:5	8.61 ± 0.46 ^c^	8.36 ± 0.28 ^a^	7.27 ± 0.24 ^bc^	6.54 ± 0.22 ^a^	8.38 ± 0.45 ^a^	60.83 ± 0.19 ^d^	3.21 ± 0.49 ^c^	1.57 ± 0.99 ^a^
70:25:5	8.45 b ± 0.45 ^c^	10.03 ± 0.34 ^b^	7.45 ± 0.25 ^bcd^	7.19 ± 0.24 ^bcd^	10.54 ± 0.57 ^bc^	56.32 ± 0.04 ^d^	3.56 ± 0.49 ^c^	2.10 ± 0.05 ^b^
75:20:5	7.37 ± 0.40 ^ab^	12.54 ± 0.43 ^c^	7.80 ± 0.26 ^cd^	7.31 ± 0.25 ^cd^	11.87 ± 0.64 ^cd^	53.09 ± 0.14 ^b^	3.71 ± 0.00 ^c^	3.25 ± 0.00 ^c^
80:15:5	7.13 ± 0.38 ^a^	12.92 ± 0.44 ^c^	7.94 ± 0.27 ^d^	7.52 ± 0.26 ^d^	12.62 ± 0.68 ^d^	51.84 ± 0.20 ^a^	3.98 ± 0.92 ^d^	3.35 ± 0.46 ^c^

Values are means ± SD of duplicate determinations. Means in the same column with different superscripts are significantly different (*p* < 0.05). P—plantain flour; B—breadfruit flour; C—cinnamon powder.

**Table 3 foods-13-00852-t003:** Functional properties of composite flour produced from plantain, breadfruit, and cinnamon (*n* = 12).

P:B:C Sample	BD g/mL	WAC %	OAC %	SP %	SI %	DIS %
100:0:0	0.78 ± 0.00 ^e^	2.28 ± 0.01 ^d^	0.88 ± 0.02 ^a^	5.30 ± 0.01 ^e^	0.02 ± 0.00 ^a^	66.05 ± 0.07 ^f^
0:100:0	0.66 ± 0.00 ^a^	2.43 ± 0.01 ^e^	1.34 ± 0.02 ^e^	3.89 ± 0.02 ^a^	0.04 ± 0.00 ^d^	48.55 ± 0.04 ^a^
65:30:5	0.74 ± 0.00 ^b^	2.87 ± 0.01 ^f^	1.03 ± 0.02 ^b^	5.12 ± 0.02 ^d^	0.03 ± 0.00 ^c^	55.49 ± 0.13 ^b^
70:25:5	0.76 ± 0.00 ^c^	1.91 ± 0.02 ^a^	1.09 ± 0.00 ^c^	4.88 ± 0.01 ^c^	0.02 ± 0.00 ^b^	57.15 ± 0.06 ^c^
75:20:5	0.78 ± 0.00 ^d^	2.00 ± 0.01 ^b^	1.16 ± 0.02 ^d^	4.69 ± 0.03 ^b^	0.02 ± 0.00 ^a^	59.54 ± 0.16 ^d^
80:15:5	0.80 ± 0.00 ^f^	2.09 ± 0.01 ^c^	1.91 ± 0.02 ^f^	3.90 ± 0.03 ^a^	0.01 ± 0.00 ^a^	60.92 ± 0.06 ^e^

Values are means ± SD of duplicate determinations. Means in the same column with different superscripts are significantly different (*p* < 0.05). BD—bulk density; WAC—water absorption capacity; OAC—oil absorption capacity; SP—swelling power; SI—solubility index; DIS—dispersibility.

**Table 4 foods-13-00852-t004:** Color attributes and anti-nutritional properties of composite flour produced from plantain, breadfruit, and cinnamon (*n* = 12).

P:B:C Sample	L*	a*	b*	∆E*
100:0:0	86.73 ± 0.35 ^ab^	2.09 ± 0.00 ^ab^	15.88 ± 0.00 ^a^	21.21 ± 0.01 ^a^
0:100:0	93.36 ± 0.49 ^b^	0.70 ± 0.03 ^a^	19.47 ± 0.21 ^c^	22.83 ± 0.28 ^b^
65:30:5	84.92 ± 0.23 ^a^	2.29 ± 0.01 ^b^	16.00 ± 0.01 ^b^	21.99 ± 0.21 ^a^
70:25:5	85.19 ± 0.07 ^ab^	2.33 ± 0.00 ^b^	16.55 ± 0.02 ^b^	22.35 ± 0.21 ^b^
75:20:5	84.63 ± 0.00 ^a^	2.42 ± 0.00 ^b^	16.75 ± 0.00 ^b^	22.84 ± 0.00 ^b^
80:15:5	85.40 ± 0.28 ^ab^	2.26 ± 0.00 ^b^	16.62 ± 0.00 ^b^	22.31 ± 0.07 ^b^

Values are means ± SD of duplicate determinations. Means in the same column with different superscripts are significantly different (*p* < 0.05).

**Table 5 foods-13-00852-t005:** Pasting properties of flours produced from plantain, breadfruit, and cinnamon (*n* = 12).

P:B:C Samples	Peak Viscosity (RVU)	Trough Viscosity (RVU)	Breakdown Viscosity (RVU)	Final Viscosity (RVU)	Setback Viscosity (RVU)	Peak time (Min)	Pasting Temperature (°C)
100:0:0	383.86 ± 1.27 ^b^	27.76 ± 1.27 ^a^	132.38 ± 2.93 ^bc^	472.25 ± 0.57 ^a^	210.99 ± 1.13 ^a^	5.16 ± 0.57 ^a^	87.04 ± 0.07 ^c^
0:100:0	331.12 ± 0.85 ^a^	313.51 ± 2.94 ^bc^	49.20 ± 1.44 ^a^	561.04 ± 0.37 ^c^	244.23 ± 0.42 ^c^	5.40 ± 0.99 ^a^	84.54 ± 0.18 ^a^
65:30:5	415.72 ± 0.99 ^c^	314.92 ± 1.13 ^bc^	724.80 ± 64.35 ^a^	570.81 ± 5.23 ^b^	240.57 ± 2.41 ^c^	5.32 ± 0.11 ^a^	85.79 ± 0.99 ^b^
70:25:5	412.79 ± 1.41 ^c^	354.89 ± 6.67 ^c^	111.41 ± 0.66 ^b^	527.52 ± 1.27 ^b^	226.25 ± 0.57 ^b^	5.17 ± 0.04 ^a^	85.28 ± 0.04 ^b^
75:20:5	443.72 ± 4.95 ^d^	297.95 ± 0.85 ^b^	162.79 ± 0.59 ^c^	560.62 ± 5.94 ^c^	248.56 ± 3.52 ^c^	5.21 ± 0.08 ^a^	86.07 ± 0.14 ^b^
80:15:5	500.98 ± 9.62 ^e^	367.87 ± 0.74 ^c^	159.17 ± 5.12 ^b^	521.81 ± 0.99 ^b^	220.39 ± 0.98 ^b^	5.38 ± 0.28 ^b^	88.24 ± 0.28 ^c^

Values are means ± SD of duplicate determinations. Mean values in the same column with different superscripts are significantly (*p* < 0.05) different.

**Table 6 foods-13-00852-t006:** Proximate composition of cookies produced from plantain–breadfruit–cinnamon flour (*n* = 24).

Samples P:B:C	Moisture Content (%)	Crude Fat (%)	Crude Ash (%)	Crude Fiber (%)	Crude Protein (%)	Carbohydrates (%)
100:0:0	7.65 ± 0.42 ^b^	18.99 ± 1.15 ^ab^	4.38 ± 0.15 ^b^	5.34 ± 0.22 ^b^	10.09 ± 0.74 ^ab^	53.58 ± 0.78 ^d^
0:100:0	6.19 ± 0.34 ^a^	17.08 ±1.04 ^ab^	3.10 ± 0.11 ^a^	3.62 ± 0.15 ^a^	10.99 ± 0.81 ^c^	59.03 ± 0.16 ^f^
65:30:5	7.99 ± 0.43 ^b^	16.06 ± 0.98 ^a^	4.26 ± 0.15 ^b^	5.52 ± 0.23 ^b^	8.80 ± 0.65 ^ab^	57.38 ± 0.18 ^e^
70:25:5	7.67 ± 0.42 ^b^	19.93 ±1.22 ^bc^	6.01 ± 0.20 ^c^	6.52 ± 0.27 ^c^	8.18 ± 0.60 ^a^	51.66 ± 0.13 ^c^
75:20:5	6.40 ± 0.35 ^a^	24.75 ± 1.51 ^d^	8.13 ± 0.28 ^e^	7.15 ± 0.30 ^d^	9.75 ± 0.72 ^abc^	43.82 ± 0.42 ^a^
80:15:5	7.00 ± 0.38 ^ab^	22.82 ± 1.40 ^cd^	7.23 ± 0.25 ^e^	7.09 ± 0.30 ^cd^	9.44 ± 0.69 ^abc^	46.44 ± 0.28 ^b^

Values are means ± SD of duplicate determinations of a duplicate sample. Mean values in the same column with different superscripts are significantly (*p* < 0.05) different.

**Table 7 foods-13-00852-t007:** Color attributes of cookies produced from plantain–breadfruit–cinnamon flour (*n* = 24).

Samples P:B:C	L*	a*	b*	ΔE*
100:0:0	27.89 ± 0.20 ^c^	4.82 ± 0.01 ^a^	13.94 ± 0.05 ^a^	15.69 ± 0.06 ^a^
0:100:0	33.28 ± 0.37 ^e^	6.10 ± 0.04 ^d^	22.75 ± 0.18 ^e^	24.87 ± 0.24 ^d^
65:30:5	26.55 ± 0.33 ^b^	5.5 ± 0.02 ^b^	15.84 ± 0.12 ^c^	17.84 ± 0.07 ^b^
70:25:5	29.37 ± 0.11 ^d^	5.49 ^b^ ± 0.11 ^b^	16.89 ± 0.18 ^d^	18.66 ± 0.20 ^c^
75:20:5	26.33 ± 0.44 ^b^	6.27 ± 0.07 ^d^	16.86 ± 0.26 ^d^	19.04 ± 0.20 ^c^
80:15:5	24.61 ± 0.54 ^a^	5.82 ± 0.09 ^c^	15.20 ± 0.45 ^b^	17.69 ± 0.31 ^b^

Values are means ± SD of duplicate determinations. Mean values in the same column with different superscripts are significantly (*p* < 0.05) different.

**Table 8 foods-13-00852-t008:** Sensory evaluation of cookies produced from plantain–breadfruit–cinnamon flour (*n* = 240).

Samples P:B:C	Color	Taste	Aroma	Crunchiness	Texture	Overall Acceptability
100:0:0	6.62 ± 1.63 ^c^	7.33 ± 1.31 ^d^	6.85 ± 1.35 ^d^	7.04 ± 1.71 ^d^	7.04 ± 1.53 ^c^	7.33 ± 1.15 ^e^
0:100:0	6.61 ± 1.19 ^c^	5.42 ± 1.28 ^a^	5.66 ± 1.68 ^a^	6.23 ± 1.64 ^b^	6.80 ± 1.74 ^b^	6.80 ± 1.74 ^b^
65:30:5	6.85 ± 1.27 ^e^	6.71 ± 1.23 ^c^	6.04 ± 1.56 ^b^	7.28 ± 1.14 ^d^	6.95 ± 1.39 ^b^	7.09 ± 1.60 ^c^
70:25:5	5.90 ± 1.30 ^a^	6.14 ± 1.62 ^b^	5.71 ± 1.76 ^a^	6.09 ± 1.30 ^a^	6.38 ± 1.16 ^a^	6.52 ± 1.32 ^a^
75:20:5	6.71 ± 1.10 ^d^	6.80 ± 1.60 ^c^	6.28 ± 1.34 ^b^	6.38 ± 1.62 ^b^	6.61 ± 1.11 ^a^	7.52 ± 0.92 ^d^
80:15:5	6.47 ± 1.77 ^b^	6.76 ± 1.70 ^c^	6.71 ± 1.27 ^c^	6.71 ± 1.64 ^c^	7.09 ± 1.26 ^c^	7.38 ± 1.35 ^d^

Values are means ± SD of duplicate determinations. Mean values in the same column with different superscripts are significantly (*p* < 0.05) different.

## Data Availability

The original contributions presented in the study are included in the article, further inquiries can be directed to the corresponding author.

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
