# Peer review of "Production and Characterization of Snacks Utilizing Composite Flour from Unripe Plantain (Musa paradisiaca), Breadfruit (Artocarpus altilis), and Cinnamon (Cinnamomum venum)"

_foods, 2024, doi:10.3390/foods13060852_

Round 1

Reviewer 1 Report

Comments and Suggestions for Authors

The present manuscript evaluated the quality of snacks produced from the composite flour of Unripe plantain, breadfruit, and cinnamon. Overall, it has some merits. However, modifications were still required to improve the whole quality of this manuscript.

1. Moderate language editing for English-writing improvement is still required.

2. Materials and methods: Strongly suggest that the first paragraph of this part should be “2.1. Materials”, which was used to list the reagents, consumable items, and special materials utilized in this study. Also, their sources, suppliers, and/or purities should be presented.

3. The "Materials and methods" section was very disordered and confused with chaotic sub-titles and structures. Strongly suggest to re-write this section.

4. Table 1: The authors didn't set up a group with P:B:C=0:0:100. Why?

5. Tables 2-7: The superscripts representing statistical significance were not marked at the same position for different Tables. Some were behind the means but others were behind the SDs. Revise them.

6. Tables 7-8: Table 7 was not a correct three-line Table. For Table 8, the significance mark for differentia data is missing.

7. References: Too many. Normally, 40±10 cited literatures are appropriate and unnecessary literatures can be removed from the list. Meanwhile, some literatures were too old, even have published for over 20 years.

Comments on the Quality of English Language

Moderate language editing is still required to improve the whole writing quality of this manuscript.

Author Response

We thank you for taking the time to review our manuscript and appreciate your comments. We carefully and thoroughly addressed all the comments and uploaded the revised manuscript, highlighting the changes using track changes.

We thank you for taking the time to review our manuscript and appreciate your comments. We carefully and thoroughly addressed all the comments and uploaded the revised manuscript, highlighting the changes using track changes.

Reviewer 1

The present manuscript evaluates the quality of snacks produced from the composite flour of unripe plantains, breadfruit, and cinnamon. Overall, it has some merits. However, modifications were still required to improve the overall quality of this manuscript.

Response: We appreciate your sincere comments about our manuscript. We have made the necessary modifications as suggested and believe the manuscript quality has improved tremendously.

  1. Moderate language editing for English-writing improvement is still required.

Response: The manuscript has undergone the language editing process as recommended. The changes made are highlighted in the track changes in the revised manuscript uploaded.

  1. Materials and methods: Strongly suggest that the first paragraph of this part should be “2.1. Materials,"  which was used to list the reagents, consumable items, and special materials utilized in this study. Also, their sources, suppliers, and/or purities should be presented.

Response: We have added the information shown below as suggested:

“The plantains, breadfruit, and cinnamon were procured from the Kuto market in Abeokuta, Ogun State, Nigeria. Also, all other ingredients, including margarine, granulated sugar, baking powder, salt, vanilla extract, powdered milk, and fresh eggs, were shopped from a local supermarket.”.

  1. The "Materials and Methods" section was very disordered and confused, with chaotic sub-titles and structures. I strongly suggest rewriting this section.

Response: We have re-written and re-structured the subtitles under materials and methods as suggested

  1. Table 1: The authors didn't set up a group with P:B:C=0:0:100. Why?

Response: We had P:B:C (100:0:0 and 0:100:0) labelled Recipes A and B in the table. Plantain and breadfruit flour were flours with starch content, and we omitted PBC (0:0:100) because it is not scientifically and logically right to use 100% cinnamon (C) alone with no starch.

  1. Tables 2–7: The superscripts representing statistical significance were not marked in the same position for different tables. Some were behind the means, but others were behind the SDs. Revise them.

Response: We have revised all the superscript letters, and they are now in the same position as highlighted in track changes in the uploaded revised manuscript.

  1. Tables 7-8: Table 7 was not a correct three-line table. For Table 8, the significance mark for differentia data is missing.

Response: We have corrected the tables as three-line tables, and the missing superscripts to mark the significant differences were added to Table 8.

  1. References: Too many. Normally, 40±10 cited pieces of literature are appropriate, and unnecessary pieces can be removed from the list. Meanwhile, some literature was too old; some had been published for over 20 years.

Response: We ensure the references cited are necessary and old ones are corrected, except for one or two methodology references.

Moderate language editing is still required to improve the overall writing quality of this manuscript.

Response: The manuscript has undergone the language editing process as recommended. The changes made are highlighted in the track changes in the revised manuscript uploaded.

Reviewer 2 Report

Comments and Suggestions for Authors

Dear authors, 

Please follow the provided indications to improve the quality of the paper. 

In the entire papers, please kindly follow the authors guidelines. The references should be placed as numbers in the Squared brackets. 

Phytate and tannins analysis chapters should be also numbered to follow the logical sequencing of the methodology. 

Colour determination method is not sufficient described. Please update and provide the name of the colorimeter. 

From the material and methods is not clear how many samples were formed and analysed, please provide n=? 

Under each table provide the number of observations. 

Lines 242,244 I don't see the point of these mention. 

The results from table 2 are roo much discussed without anything concrete, please try to provide comprehensive information compared with literature data. Also, maybe there is a correlation between low moisture and high fat content in the last two samples. 

Table 3 should be presented before table 2, first the chemical composition of the studied products and after their combinations. This sounds more logical. 

The tannins and phytate content should be presented different from the colorimetric parameters. 

Since the values are already presented in the tables, please do not repeat them in the text. Avoid redundancy of the data.

Table 5 is too crowded and hard to follow. 

Table 8 presented sensory evaluation,  however in the material and methods is no word about this determination and how was conducted. 

The conclusion is also too longe. 

Overall in the results, there is too much redundancy of the data  and lack of scientific explanation, only some basic comparison with previous studies, and this is not sufficient. 

Try to argument the findings from your point of view. 

Author Response

We thank you for taking the time to review our manuscript and appreciate your comments. We carefully and thoroughly addressed all the comments and uploaded the revised manuscript, highlighting the changes using track changes.

Reviewer 2

1. Please follow the provided indications to improve the quality of the paper. 

In the entire papers, please kindly follow the authors guidelines. The references should be placed as numbers in the Squared brackets.

 Response: We appreciate your sincere comments about our manuscript. We have made the necessary modifications as suggested.

2. Phytate and tannins analysis chapters should be also numbered to follow the logical sequencing of the methodology. 

Response:  The phytate and tannins analysis sections have been numbered as advised.

3. Colour determination method is not sufficient described. Please update and provide the name of the colorimeter. 

Response:  Additional information has been added to the colour measurements as suggested.

4. From the material and methods is not clear how many samples were formed and analysed, please provide n=? 

Response: The number of composite flour samples formed, and cookies produced has been stated, which indicates all analyses were duplicated at the laboratory.

5. Under each Table provide the number of observations. 

Response: The value of “n” has been added to each Table

6. Lines 242,244 I don't see the point of these mention. 

Response: These sentences have been rephrased

7. The results from table 2 are roo much discussed without anything concrete, please try to provide comprehensive information compared with literature data. Also, maybe there is a correlation between low moisture and high fat content in the last two samples.

 Response: The results from Table 2 have been rewritten, and information with the appropriate literature data was added.

8. Table 3 should be presented before table 2, first the chemical composition of the studied products and after their combinations. This sounds more logical. 

Response: We appreciate your suggestion, but we thought presenting chemical properties before functional properties was okay.

9. The tannins and phytate content should be presented different from the colorimetric parameters.

Response: We have merged it with the proximate composition Table to prevent adding a Table to the eight Tables in the manuscript.  

10. Since the values are already presented in the tables, please do not repeat them in the text. Avoid redundancy of the data.

Response: We appreciate your sincere comments about our manuscript. We have made the necessary modifications as suggested.

11. Table 5 is too crowded and hard to follow. 

Response: This is a standard way to present all the pasting properties; these are properties produced by the Rapid ViscoAnalyzer. Most papers on pasting properties presented their data in this manner.

12. Table 8 presented sensory evaluation, however, in the material and methods is no word about this determination and how was conducted. 

Response: A section on sensory evaluation has been provided under materials and methods.

13. The conclusion is also too longe. 

Response: The conclusion has been rephrased.

14. Overall, in the results, there is too much redundancy of the data and lack of scientific explanation, only some basic comparison with previous studies, and this is not sufficient. 

Try to argument the findings from your point of view. 

Response: We appreciate your sincere comments and have made the necessary modifications as suggested. All the responses are shown in Track Changes.

Round 2

Reviewer 1 Report

Comments and Suggestions for Authors

As the authors have modified the manuscript point-by-point according to the reviewers' comments and suggestions, I think it can be acceptable for the publication on this great journal in the current form.

Reviewer 2 Report

Comments and Suggestions for Authors

I am a bit confused about the way authors placed references in the text,  but I will let this to the Editorial board. 

No further observations